# Bromodomain inhibition of the transcriptional coactivators CBP/EP300 as a therapeutic strategy to target the IRF4 network in multiple myeloma

Andrew R Conery[1], Richard C Centore[1], Adrianne Neiss[1], Patricia J Keller[1], Shivangi Joshi[1], Kerry L Spillane[1], Peter Sandy[1], Charlie Hatton[1], Eneida Pardo[1], Laura Zawadzke[1], Archana Bommi-Reddy[1], Karen E Gascoigne[2], Barbara M Bryant[1], Jennifer A Mertz[1], Robert J Sims III[1]*

[1]Constellation Pharmaceuticals, Cambridge, United States; [2]Genentech, South San Francisco, United States

*For correspondence: robert.
sims@constellationpharma.com

**Abstract** Pharmacological inhibition of chromatin co-regulatory factors represents a clinically validated strategy to modulate oncogenic signaling through selective attenuation of gene expression. Here, we demonstrate that CBP/EP300 bromodomain inhibition preferentially abrogates the viability of multiple myeloma cell lines. Selective targeting of multiple myeloma cell lines through CBP/EP300 bromodomain inhibition is the result of direct transcriptional suppression of the lymphocyte-specific transcription factor IRF4, which is essential for the viability of myeloma cells, and the concomitant repression of the IRF4 target gene *c-MYC*. Ectopic expression of either IRF4 or MYC antagonizes the phenotypic and transcriptional effects of CBP/EP300 bromodomain inhibition, highlighting the IRF4/MYC axis as a key component of its mechanism of action. These findings suggest that CBP/EP300 bromodomain inhibition represents a viable therapeutic strategy for targeting multiple myeloma and other lymphoid malignancies dependent on the IRF4 network.

## Introduction

Multiple myeloma is an aggressive and incurable hematologic malignancy characterized by the proliferation of abnormal plasma cells (*Mahindra et al., 2010*). Myeloma is driven by transcriptional reprogramming events that prevent the differentiation of activated B cells to plasma cells and subsequently promote the proliferation of dysfunctional plasma cells (*Mahindra et al., 2010*). Abnormal activity of a number of transcription factors has been implicated in multiple myeloma development, including NF-κB, MAF, MYC, and interferon regulatory factor 4 (IRF4) (*Dean et al., 1983*; *Keats et al., 2007*; *Palumbo et al., 1989*; *Shaffer et al., 2008*). The oncogenic activity of these transcription factors in multiple myeloma is demonstrated by the presence of translocation events that fuse them to highly active enhancers that drive high expression (*Dean et al., 1983*; *Iida et al., 1997*).

The IRF4 transcription factor is a critical component of the normal adaptive immune response and is required for lymphocyte activation and differentiation of immunoglobulin-secreting plasma cells (*Klein et al., 2006*; *Mittrücker et al., 1997*; *Sciammas et al., 2006*). Downstream targets of IRF4 include factors that regulate cell cycle progression, survival, and normal plasma cell function (*Shaffer et al., 2008*). While oncogenic translocations of *IRF4* have been found, more frequently, myeloma and other lymphoid malignancies are dependent on dysfunctional transcriptional networks downstream of a genetically normal *IRF4* locus (*Shaffer et al., 2008*). While the immunomodulatory

**eLife digest** Multiple myeloma is an aggressive and incurable cancer of white blood cells called B cells and plasma cells. Many of the mutations that trigger multiple myeloma interfere with genes that normally cause B cells to develop into plasma cells.

Multiple myeloma cells often activate genes that are inactive in healthy cells or vice versa. They also express some genes that are active in healthy cells but at the wrong levels. These changes in gene expression are regulated by proteins that bind to DNA and other DNA-associated proteins. Proteins called CBP and EP300 are two examples of regulatory proteins, and have been implicated in promoting various cancers in humans. Both CBP and EP300 contain a region known as a bromodomain, which binds to proteins associated with DNA. Abnormal activity of the bromodomains of CBP and EP300 may thus promote the onset of cancer.

Conery et al. have now treated a wide range of human blood cancer cells grown in the laboratory with two new chemicals that inhibit CBP and EP300 bromodomains. Of all the cells tested, multiple myeloma cells were the most strongly affected; these cells proliferated more slowly and died more quickly in the presence of the chemical inhibitors.

Next, Conery et al. analysed the changes in gene expression in the multiple myeloma cells when they were treated with the inhibitors. The genes whose expression levels changed the most were genes that are regulated by a protein called IRF4. This protein is important for normal B cell and plasma cell development. One notable IRF4 target gene that was down-regulated was the gene that encodes a protein called Myc, which strongly encourages cell division and growth. Conery et al. then supplemented the multiple myeloma cells with extra IRF4 or Myc while treating with the inhibitors and found that this caused the inhibitors to lose most of their effect.

Neither CBP nor EP300 have previously been thought of as targets for multiple myeloma therapy. Therefore a next critical step is to find more refined chemicals to target their bromodomains and importantly to test these chemicals in preclinical trials. These studies could in turn lead to improved treatments for patients with multiple myeloma in the future.

agent lenalidomide has been shown to promote IRF4 protein degradation (*Moros et al., 2014*), pharmacological agents that regulate the expression of *IRF4* mRNA have not been identified.

Small molecule inhibition of bromodomain-containing transcriptional co-regulators have recently been shown to be a viable strategy for the suppression of otherwise un-druggable downstream transcription factors. This is best exemplified by the inhibitors of BET family bromodomains, which down-regulate *MYC* and *BCL2* and are thus highly active in malignancies driven by these critical oncogenes (*Dawson et al., 2011*; *Delmore et al., 2011*; *Mertz et al., 2011*; *Zuber et al., 2011*). Cyclic AMP response element binding protein (CREB)-binding protein (CBP) and E1A interacting protein of 300 kDa (EP300) are highly homologous bromodomain-containing transcriptional co-activators that regulate a number of important cellular events through their acetyltransferase activity (*Goodman and Smolik, 2000*). Genetic studies in mice and surveys of human cancer mutations and translocations have implicated CBP/EP300 in cancer, but the role of the bromodomain in the normal and pathological function of CBP/EP300 has not been extensively studied (*Kung et al., 2000*; *Murati et al., 2007*; *Ohnishi et al., 2008*; *Pasqualucci et al., 2011*; *Peifer et al., 2012*). Given the importance of these genes in cancer development, CBP/EP300 bromodomain inhibition may represent an important therapeutic strategy to reprogram oncogenic signaling pathways in human malignancies.

## Results

### Cellular specificity of CBP/EP300 bromodomain inhibitors

To assess the functional role of CBP/EP300 bromodomains, we made use of two chemical probes recently generated by the Structural Genomics Consortium (*Figure 1A*) (SGC; www.thesgc. org) (*Hay et al., 2014*). SGC-CBP30 and I-CBP112 are chemically distinct tool compounds with selective affinity for the bromodomains of CBP/EP300 over other bromodomains in this protein

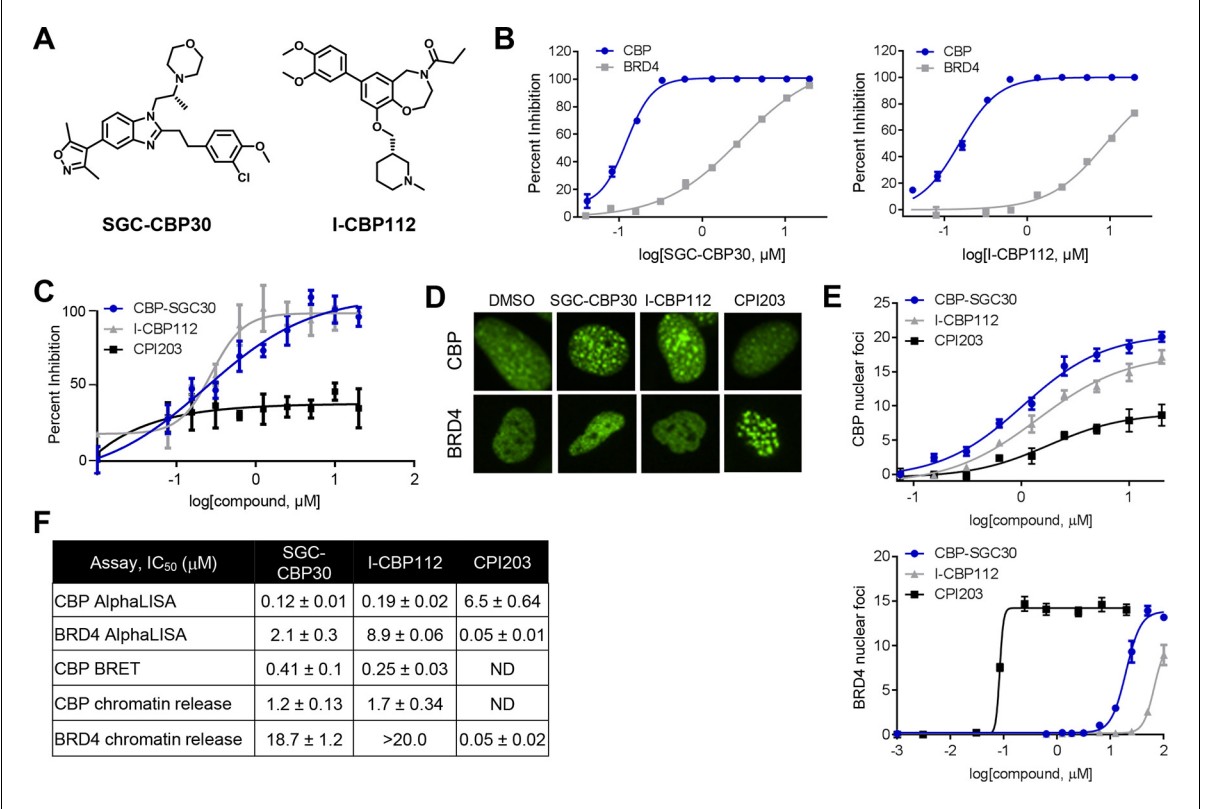

**Figure 1.** Characterization of CBP/EP300 bromodomain inhibitors. (**A**) Structures of SGC-CBP30 and I-CBP112. (**B**) Representative AlphaLISA curves showing the inhibition of acetylated peptide binding to isolated CBP or BRD4 bromodomains in the presence of SGC-CBP30 and I-CBP112. Error bars represent SEM of 3 technical replicates. (**C**) Dose-titrations of SGC-CBP30, I-CBP112, and CPI203 using NanoBRET with the isolated CBP bromodomain and histone H3.3 in 293 cells. Error bars represent SEM of three technical replicates. The calculated EC$_{50}$ values are shown in F. (**D**) ZsGreen-bromodomain fusion proteins were monitored by high content imaging. Representative nuclei showing nuclear foci in the indicated assays in the presence of DMSO, SGC-CBP30 (5 µM), I-CBP112 (5 µM) or CPI203 (0.33 µM). (**E**) Quantification of chromatin release assay. Each curve represents a titration of the indicated compound in stable cell lines expressing the indicated fusion protein (CBP: CBP-bromodomain/BRD9; BRD4: full length BRD4). Values are mean of four fields per well of two technical replicates, ± SEM. (**F**) Summary of biochemical and cellular activity of the indicated compounds. Values represent half-maximal inhibition (IC$_{50}$) in AlphaLISA assays (n ≥ 2 independent replicates) or half-maximal induction (EC$_{50}$) in NanoBRET (n = 3 technical replicates ± SEM) or chromatin release assays (n = 2 biological replicates ± SEM). ND = <u>n</u>ot <u>d</u>etermined due to a failure to produce 100% inhibition compared to controls.

The following source data is available for figure 1:

**Source data 1.** Bromodomain selectivity of CBP/EP300 bromodomain inhibitors.

family. Independent of CBP/EP300, the bromodomains with the highest affinity for these molecules is the BET bromodomain family (*Hay et al., 2014*). We confirmed the biochemical potency and selectivity of SGC-CBP30 and I-CBP112 using AlphaLISA with the isolated bromodomain of CBP and the first bromodomain of BRD4 (BRD4-BD1) (*Figure 1B,F*). We further addressed the selectivity of the compounds through the use of Differential Scanning Fluorimetry (DSF) with a panel of 19 purified bromodomains (*Figure 1—source data 1*). Taken together, these data are consistent with published reports regarding the selectivity of these compounds (*Hammitzsch et al., 2015; Picaud et al., 2015*).

To assess the potency of these probes in cells, we utilized a proximity-based assay (NanoBRET), which monitors the interaction between the bromodomain of CBP and histone H3.3. SGC-CBP30 and I-CBP112 showed similar dose-dependent inhibition of CBP-H3.3 binding, with calculated EC$_{50}$ values of 0.28 µM and 0.24 µM, respectively (*Figure 1C and F*). The BET bromodomain inhibitor

CPI203 (*Devaiah et al., 2012*) did not display dose-dependent inhibition in this assay (*Figure 1C*). Next, we made use of an imaging-based assay that measures the release of bromodomain-GFP fusion proteins from chromatin upon ligand binding (*Huang et al., 2014*). As shown in *Figure 1D*, chromatin release results in aggregation of fusion proteins into finite speckles whose number and intensity increase with ligand binding. Both SGC-CBP30 and I-CBP112 promote chromatin release of CBP bromodomain fusion proteins at low micromolar concentrations as quantitated by high-content imaging (10-fold cell shift), comparable to previous results (*Figure 1E,F*) (*Hay et al., 2014*). In contrast, both probe compounds release BRD4-BD1 fusion proteins from chromatin at significantly higher concentrations as compared to the selective BET inhibitor CPI203 (*Figure 1D–F*) (*Devaiah et al., 2012*). Given the cellular selectivity of the compounds, we are confident that at defined concentrations of the inhibitors (≤2.5 µM SGC-CBP30 or ≤5 µM I-CBP112), any observed pharmacological effects are due to on-target inhibition of CBP/EP300 bromodomains.

## CBP/EP300 bromodomain inhibition causes cell cycle arrest and apoptosis

To assess the biological activity of CBP/EP300 bromodomain inhibition, we treated a panel of cell lines of multiple myeloma and acute leukemia origin with SGC-CBP30 and I-CBP112. As shown in *Figure 2A,B*, and *Figure 2—figure supplement 1A*, a subset of cell lines was highly sensitive to both compounds, with the most sensitive cell lines having $GI_{50}$ values below 3 µM of SGC-CBP30. Notably, 14 of the 15 most sensitive cell lines are of multiple myeloma origin (*Figure 2A*). As effectors of multiple biological processes, CBP and EP300 play important roles in multiple phases of the cell cycle. To assess the requirement of the CBP/EP300 bromodomains in cell cycle progression, we released G0/G1 arrested LP-1 cells in the presence of either DMSO, SGC-CBP30, or I-CBP112. As shown in *Figures 2C* and *Figure 2—figure supplement 1B*, the progression of the cells appears normal through G2/M phase (8 hr). Only upon entry into the next cell cycle is there a noticeable alteration in cell cycle progression, with compound-treated cells accumulating in G1 at 16 and 24 hr as compared to DMSO-treated cells. Thus, it appears that the primary phenotypic effect of CBP/EP300 bromodomain inhibition is arrest in the G1 phase of the cell cycle. Consistent with these observations, growth inhibition resulting from CBP/EP300 bromodomain inhibition is accompanied by G0/G1 arrest and apoptosis in phenotypically sensitive cell lines (*Figures 2D* and *Figure 2—figure supplement 1C*). As the phenotypic effects of SGC-CBP30 and I-CBP112 appeared similar, we utilized the more potent compound, SGC-CBP30, for further experiments and made use of I-CBP112 as a distinct chemotype to confirm important observations.

## CBP/EP300 bromodomain inhibition targets the IRF4 transcriptional program

Recent work by many groups has demonstrated that small molecule inhibitors of BET family bromodomains are highly active in cell lines of hematopoetic origin (*Dawson et al., 2011*; *Delmore et al., 2011*; *Mertz et al., 2011*; *Zuber et al., 2011*). In contrast, our results suggest that CBP/EP300 bromodomain inhibition preferentially targets a more limited subset of hematologic cell lines, with a bias toward multiple myeloma/plasmacytoma cell lines. To gain insight into the mechanisms underlying these phenotypic differences, we carried out RNA sequencing of LP-1 cells treated with SGC-CBP30 or the pan-BET inhibitor CPI203. To narrow our focus to direct transcriptional effects, we examined gene expression changes following short term (6 hr) compound treatment. As shown in *Figures 3A* and *Figure 3—figure supplement 1A*, the transcriptional footprint of SGC-CBP30 is more circumscribed than that of CPI203, with far fewer genes differentially expressed. Notably however, the genes differentially expressed by SGC-CBP30 are not simply a subset of those affected by CPI203 (*Figures 3A* and *Figure 3—figure supplement 1A*; confirmed with I-CBP112 in *Figure 3—figure supplement 1D*). This suggests that the two modalities may target distinct transcriptional pathways.

To better understand the pathways impacted by CBP/EP300 and BET bromodomain inhibition, we carried out Gene Set Enrichment Analysis (GSEA) (*Subramanian et al., 2005*) with an emphasis on transcriptional pathways that might distinguish the two modalities. As expected, gene signatures negatively correlated with CPI203 treatment were dominated by MYC-dependent transcriptional pathways (*Figure 3—figure supplement 1E*). However, while several MYC signatures were also

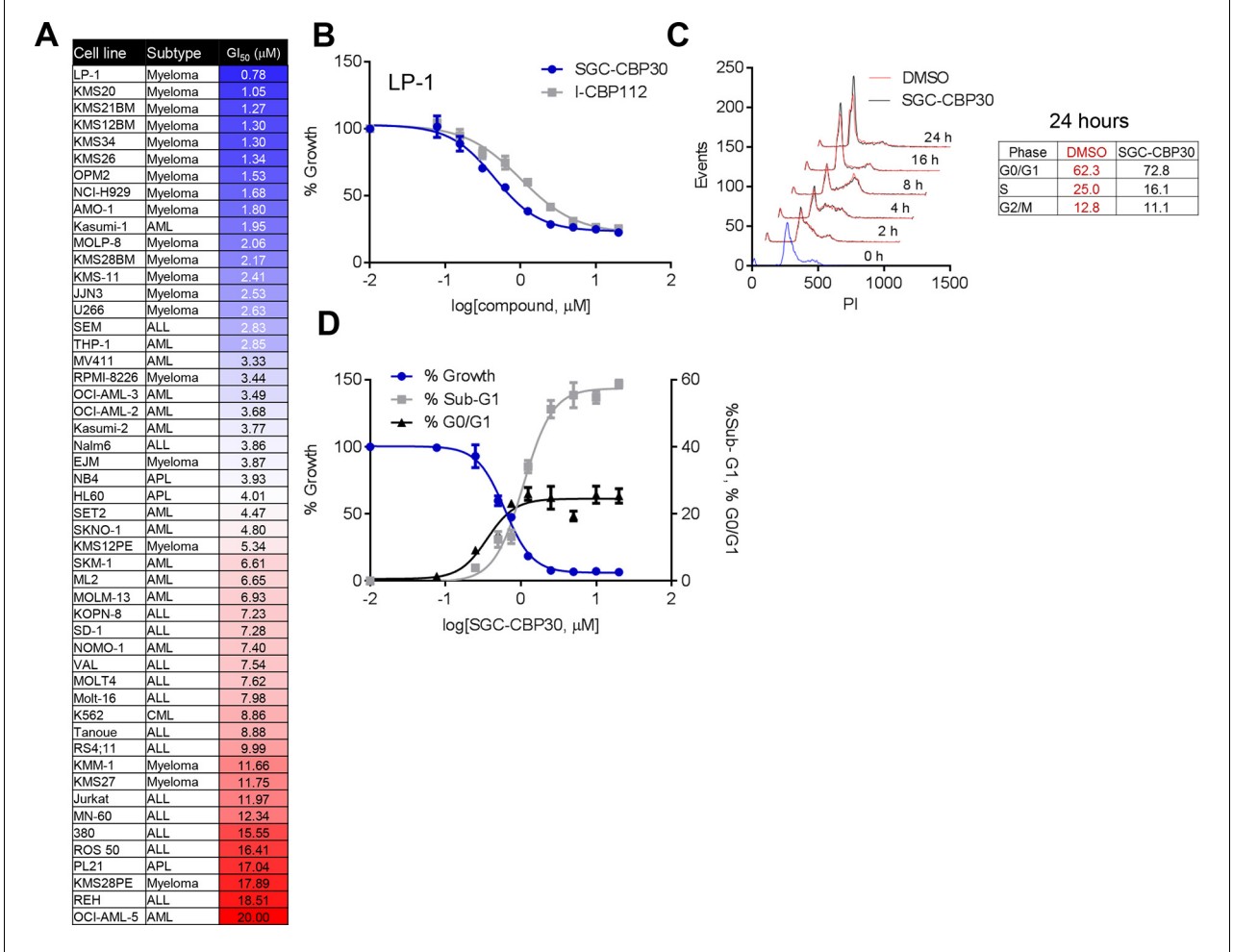

**Figure 2.** Phenotypic effects of CBP/EP300 bromodomain inhibition. (A) Growth inhibitory effects of SGC-CBP30 and I-CBP112 in the indicated cell lines. Cells were incubated with compounds for 6 days, and viability was measured with resazurin. Values are the mean of at least two biological replicates. Values with error can be found in *Figure 2—source data 1*. (B) Example viability curves for LP-1. Values represent the mean of three3 technical replicates, ± SD. (C) LP-1 were synchronized by double thymidine block and released into either DMSO or 2.5 μM SGC-CBP30. Cells were fixed and stained with PI for cell cycle analysis at the indicated time points. Cell cycle distribution at 24 hr is shown in the table. Representative data from one of two biological replicates are shown. (D) LP-1 cells were treated as in (A) and fixed after 6 days. Viable cell number and percent increase in G0/G1 or sub-G1 over DMSO were determined by flow cytometry. Each point is the mean of three technical replicates, ± SD. See *Figure 2—figure supplement 1* for additional data with I-CBP112.

The following source data and figure supplement are available for figure 2:

**Source data 1.** GI50 and standard deviation for a minimum of two replicates for the data shown in *Figure 2* and *Figure 2—figure supplement 1*.

**Figure supplement 1.** CBP/EP300 bromodomain inhibition affects the viability of multiple myeloma cells.

enriched upon treatment with SGC-CBP30, more notable was the enrichment of signatures for pathways important in multiple myeloma (*Figures 3B and C*), which was distinct from the effects of BET inhibition. We noted in particular the significant negative correlation (p-value < 0.05) of 4 gene signatures containing downstream targets of IRF4, a lymphocyte-specific transcription factor that is essential for the survival of multiple myeloma cells (*Figure 3—figure supplement 1B*) (*Shaffer et al., 2008*). Consistent with this gene set enrichment, IRF4 target genes (catalogued by Shaffer et al.) are significantly enriched in the set of genes differentially expressed following treatment with SGC-CBP30 (*Figure 3—figure supplement 1C*). A subset of these IRF4 target genes

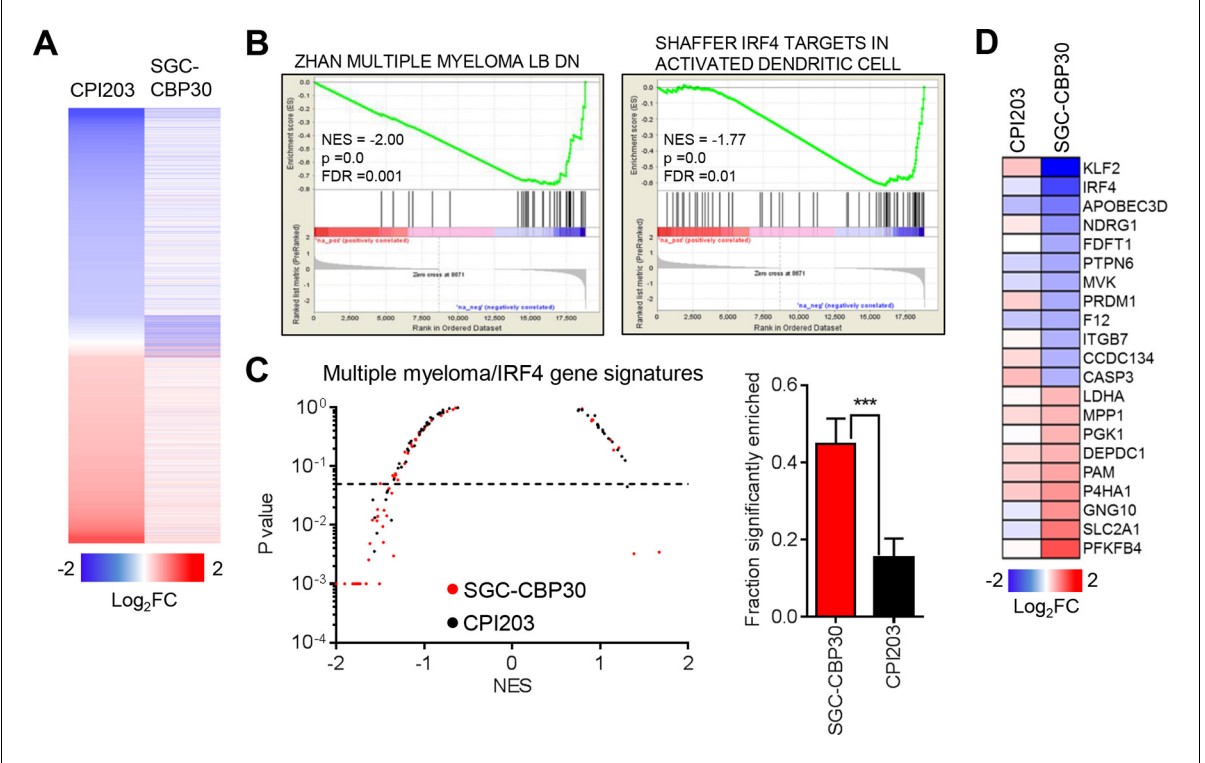

**Figure 3.** CBP/EP300 bromodomain inhibition targets IRF4. (A) LP-1 cells were treated with SGC-CBP30 (2.5 µM) or CPI203 (0.25 µM) for 6 hr, and mRNA expression was measured using RNA sequencing. Expression values for biological replicate compound-treated samples were normalized to paired DMSO controls to obtain $log_2$ fold change values. (B) Example enrichment plots for GSEA of SGC-CBP30 treated LP-1 cells. (C) Left, Scatter plot of P value vs. NES for multiple myeloma and IRF4 gene signatures for SGC-CBP30 (red) or CPI203 (black) treated LP-1 cells. Dashed line indicates p = 0.05. Right, fraction of gene signatures significantly enriched (p<0.05) with each treatment. Error bars indicate SEM. SGC-CBP30: 26/58; CPI203: 9/58. *** indicates p = 0.0005 by unpaired 2-tailed t-test. (D) IRF4 target genes differentially expressed (minimum 1.5 fold, p<0.05) with SGC-CBP30, but not CPI203. See *Figure 3—figure supplement 1* for additional gene expression data and analysis.

The following figure supplement is available for figure 3:

**Figure supplement 1.** CBP/EP300 bromodomain inhibition targets IRF4 transcriptional programs.

(including *IRF4* itself) is significantly differentially expressed following treatment with SGC-CBP30 but not CPI203 (*Figure 3D*), arguing that the IRF4 transcriptional axis may be selectively targeted by CBP/EP300 bromodomain inhibition.

## CBP/EP300 bromodomain inhibition directly suppresses the expression of *IRF4*

Since the regulation of the IRF4 transcriptional axis through small molecule inhibition of CBP/EP300 bromodomains would represent a promising new therapeutic strategy for multiple myeloma, we sought to better understand our initial observations. We first demonstrated by qRT-PCR that *IRF4* mRNA was suppressed in a dose-dependent manner by CBP/EP300 bromodomain inhibition in both LP-1 and another multiple myeloma cell line, OPM2 (*Figure 4A* and *Figure 4—figure supplement 1A*). The $EC_{50}$ of *IRF4* suppression in each cell line is in the range of the cellular $EC_{50}$ values shown in *Figure 1* and the $GI_{50}$ values shown in *Figure 2A*, arguing for an on-target effect. Consistent with suppression at the mRNA level, IRF4 protein is reduced upon treatment with SGC-CBP30 or I-CBP112 (*Figure 4—figure supplement 1D*). In support of a direct effect on the transcription of *IRF4*, we observed that *IRF4* is suppressed within 2 hr of addition of SGC-CBP30 (*Figure 4—figure supplement 1C*), and recovers within 1 hr of removal of the compound (*Figure 4B*).

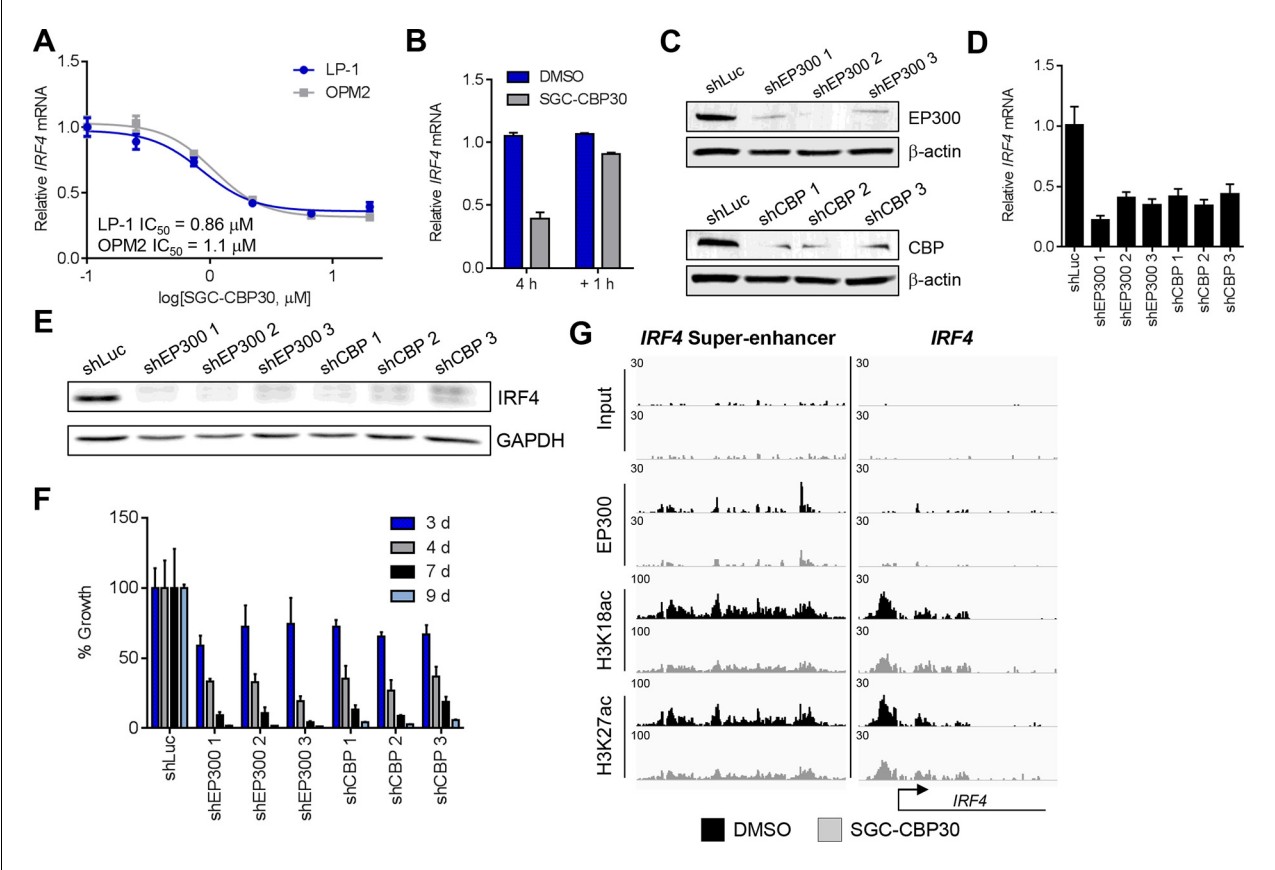

**Figure 4.** *IRF4* is a direct transcriptional target of CBP/EP300 bromodomain inhibition. (**A**) Dose-dependent inhibition of *IRF4* mRNA expression (qRT-PCR) with SGC-CBP30 in LP-1 and OPM2 cells following 6 hr of treatment. Values represent the mean of three biological replicates, ± SEM. (**B**) LP-1 cells were treated with SGC-CBP30 (2.5 μM) for 4 hr, compound was removed, and cells were incubated for an additional 1 hr in fresh media. Levels of *IRF4* mRNA were measured by qRT-PCR and normalized to *GAPDH*. Relative mRNA values normalized to DMSO at each time point represent the mean of 2 biological replicates, ± SEM. (**C**) Cells were transduced with lentivirus and lysed for Western analysis with the indicated antibodies (3 days post-infection). (**D**) *IRF4* expression was determined by qRT-PCR at 3.5 days following the transduction of shRNA lentivirus, and mRNA was normalized to *GAPDH* and expressed relative to the control shLuc (n = 3 technical replicates, ± SEM). (**E**) Western analysis with the indicated antibodies was carried out at 3.5 days post-transduction with the indicated shRNA constructs. (**F**) Cells were fixed at the indicated time points following transduction and viability was determined by flow cytometry. Percent growth is expressed relative to control shLuc at each time point. Values represent the mean of n = 3 technical replicates, ± SEM. (**G**) LP-1 cells were treated with SGC-CBP30 (2.5 μM) for 6 hr, and the indicated antibodies were used for ChIP-seq. Sequencing traces for the IRF4 super-enhancer and the transcriptional start site are shown. See *Figure 4—figure supplements 1* and *2* for additional supporting data.

The following figure supplements are available for figure 4:

**Figure supplement 1.** CBP/EP300 bromodomain inhibition regulates the expression of IRF4.

**Figure supplement 2.** CBP/EP300 bromodomain inhibition does not cause global eviction of BRD4 from chromatin.

To further corroborate that *IRF4* suppression is due to the on-target activity of CBP/EP300, we used RNAi to knock down either CBP or EP300 in the LP-1 cell line. As shown in *Figures 4C,D and E*, three unique shRNA constructs that efficiently knocked down either CBP or EP300 reduced the expression of *IRF4* at the mRNA and protein level. Viability effects were observed subsequent to suppression of *IRF4* (*Figure 4F*), which is consistent with the kinetics and phenotypic effects of CBP/EP300 bromodomain inhibition. Taken together, these data argue that the suppression of *IRF4* is due to on target inhibition of the CBP/EP300 bromodomains.

CBP and EP300 function as transcriptional co-activators via acetylation of histones and transcription factors. The bromodomains of CBP/EP300 are required for the acetylation of histones within a chromatin context, and histone H3 lysine 18 (H3K18) and histone H3 lysine 27 (H3K27) have been shown to be specifically targeted by CBP/EP300 (*Jin et al., 2011*). To investigate the mechanism of transcriptional suppression of *IRF4*, we first examined whether CBP/EP300 bromodomain inhibition causes global reduction in histone acetylation. Following incubation of LP-1 cells with SGC-CBP30, we did not observe any significant changes in the global levels of H3K18 or H3K27 acetylation by Western analysis (*Figure 4—figure supplement 1E*). We looked more closely for localized changes in histone acetylation by using chromatin immunoprecipitation followed by massively parallel sequencing (ChIP-seq). As shown in *Figure 4G*, we observed a significant reduction in both H3K18ac and H3K27ac at a previously annotated super-enhancer of IRF4 (*Chapuy et al., 2013*) as well as at the transcription start site. Notably, this reduction in acetylation is accompanied by a reduction in the chromatin occupancy of EP300, suggesting that CBP/EP300 bromodomain inhibition promotes release of the protein from chromatin leading to a reduction in histone acetylation. It should be noted that broad and complete loss of EP300 was not observed, perhaps suggesting that the bromodomain of EP300 serves to localize it to restricted domains (*Figure 4G*). Importantly, treatment with SGC-CBP30 did not result in global eviction of BRD4, arguing against a direct effect on BET bromodomain proteins (*Figure 4—figure supplement 2*).

## IRF4 and MYC suppression are associated with phenotypic response to CBP/EP300 bromodomain inhibition

We have shown that CBP/EP300 bromodomain inhibition leads to viability defects in multiple myeloma cell lines and to the suppression of IRF4 and its downstream transcriptional programs in the representative cell line LP-1. To understand whether the suppression of IRF4 was more broadly involved in the phenotypic response to CBP/EP300 bromodomain inhibition, we profiled the transcriptional response of a panel of cell lines of varying sensitivity to SGC-CBP30 (*Figure 2A*) following a 6-hr treatment with the inhibitor. As shown in *Figure 5A*, the degree of suppression of *IRF4* mRNA is significantly correlated with phenotypic sensitivity to SGC-CBP30, suggesting that this pharmacodynamic response is important for the mechanism of growth inhibition.

To better understand the events downstream of IRF4 suppression that are important for reducing proliferation and viability following CBP/EP300 bromodomain inhibition, we reduced the expression of IRF4 in a panel of multiple myeloma cell lines through shRNA transduction. The results first indicate that those cell lines that are sensitive to SGC-CBP30 ($GI_{50} < 2.5$ µM) require IRF4 for viability (*Figures 5B* and *Figure 5—figure supplement 1A*). Further, consistent with published results (*Shaffer et al., 2008*), knockdown of IRF4 and reduction in viability are associated with concomitant suppression of the oncogenic transcription factor c-MYC (MYC).

We reasoned that CBP/EP300 bromodomain inhibition may exert its phenotypic effects through the suppression of MYC downstream of IRF4 in multiple myeloma cells. While not among the most downregulated genes following the treatment of LP-1 cells with SGC-CBP30, *MYC* expression was significantly reduced (see below), and MYC transcriptional programs were affected (*Figure 3—figure supplement 1E*). Further, as with *IRF4* suppression, the degree of suppression of *MYC* mRNA in a panel of cell lines is significantly correlated with phenotypic sensitivity to SGC-CBP30 (*Figure 5—figure supplement 1B*). To confirm the dose-dependent reduction of *MYC* expression, we treated LP-1 and OPM2 cells with either SGC-CBP30 or I-CBP112 (*Figures 5C* and *Figure 5—figure supplement 1C*). The expression of *MYC* was reduced in a dose-dependent manner, with IC50 values somewhat higher than those observed for *IRF4* suppression (*Figures 4A* and *Figure 4—figure supplement 1A*). We also noted that H3K18ac and H3K27ac were reduced at the chromatin regions driving *MYC* expression following CBP/EP300 bromodomain inhibition, although loss of EP300 was less apparent, consistent with *IRF4* suppression being up-stream of *MYC* suppression in this context (*Figure 5—figure supplement 1D*). Further, consistent with the suppression of IRF4 at both the mRNA and protein levels (*Figures 4D,E*), the expression of MYC was reduced following the knockdown of either EP300 or CBP in LP-1 and OPM2 cells (*Figure 5D*, *Figure 5—figure supplement 1E and F*). Taken together, these data suggest that the bromodomains of CBP and EP300 are involved in the regulation of the IRF4/MYC axis in multiple myeloma cells, and the suppression of the IRF4/MYC axis may be important for the phenotypic effects of CBP/EP300 bromodomain inhibition.

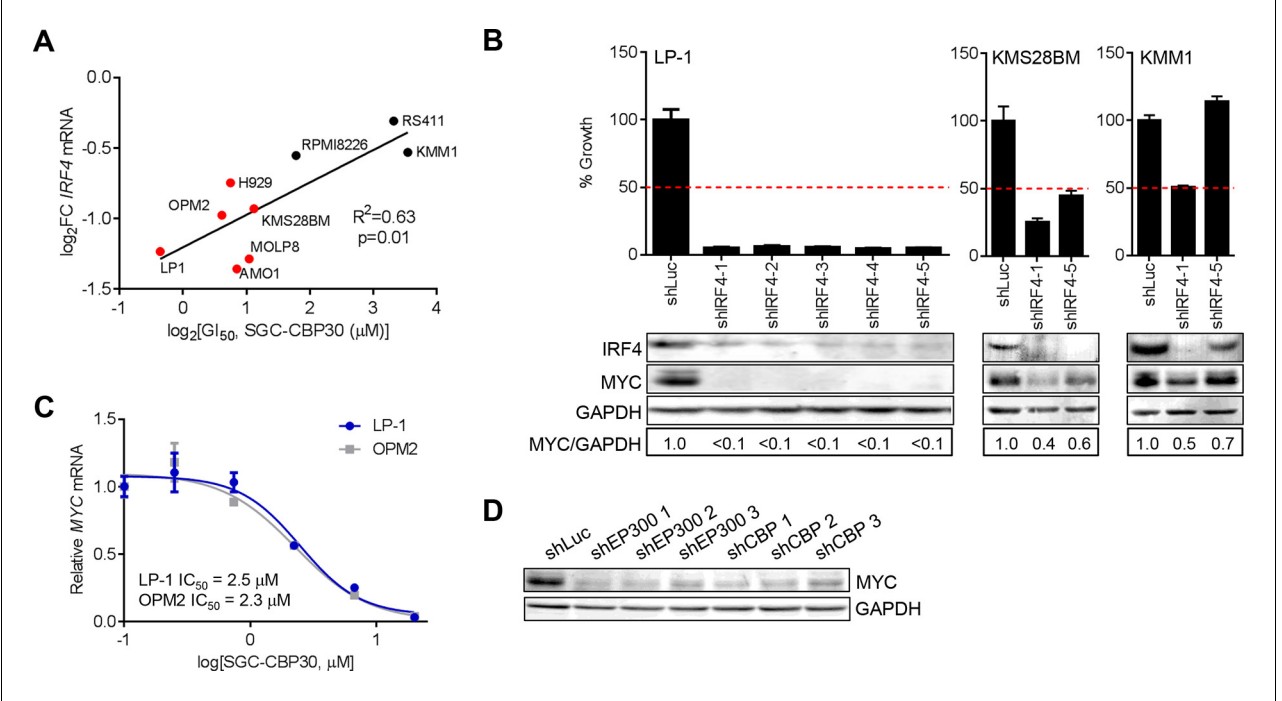

**Figure 5.** *IRF4* suppression is correlated with phenotypic sensitivity to SGC-CBP30, and MYC is downregulated concomitant with *IRF4* suppression following CBP/EP300 knockdown or bromodomain inhibition. (**A**) The indicated cell lines were treated with SGC-CBP30 (2.5 μM) for 6 hr, and *IRF4* expression normalized to *GAPDH* was determined by q-RTPCR. Suppression of *IRF4* ($\log_2$ fold change relative to DMSO) was plotted against $\log_2$ GI$_{50}$. $R^2$ and p-value of the linear regression are shown. Cell lines indicated in red have a GI$_{50}$ of less than 2.5 μM SGC-CBP30 (*Figure 2A*). Source data can be found in *Figure 5—source data 1*. (**B**) Lentiviral shRNA constructs were transduced into the indicated cell lines. Western analysis was carried out after 4 days, and viability (n = 3 technical replicates ± SEM), was assessed after 7 days. Intensity of MYC bands relative to GAPDH bands is shown below the Western blots. (**C**) Cells were treated as in *Figure 4A*, and normalized expression of *MYC* was determined by q-RTPCR. Values represent the mean of three biological replicates, ± SEM. (**D**) LP-1 cells were transduced as in *Figure 4E*, and MYC protein expression was determined by Western analysis. See *Figure 5—figure supplements 1* and *Figure 5—source data 1* for additional data.

The following source data and figure supplement are available for figure 5:

**Source data 1.** Source data for *Figure 5A* and *Figure 5—figure supplement 1B*.

**Figure supplement 1.** Suppression of the IRF4/MYC axis is important for the effects of CBP/EP300 bromodomain inhibition.

## Suppression of the IRF4/MYC axis is required for anti-myeloma effects of CBP/EP300 bromodomain inhibition

To further test the link between the transcriptional effects on IRF4/MYC and the phenotypic consequences of CBP/EP300 bromodomain inhibition, we generated LP-1 cell lines containing inducible IRF4 (LP-1/IRF4) or MYC (LP-1/MYC) expression cassettes. We then treated these cell lines with SGC-CBP30 or I-CBP112 in the presence or absence of doxycycline to induce ectopic expression of IRF4 or MYC. As shown in *Figure 6A* and *Figure 6—figure supplement 1*, in the absence of doxycycline, CBP/EP300 bromodomain inhibition induces G0/G1 arrest within 24 hr, consistent with our previous observations (*Figure 2C*). However, upon ectopic expression of IRF4, the cell cycle arrest is completely abrogated, indicating that suppression of *IRF4* is required for the most proximal phenotypic consequence of CBP/EP300 bromodomain inhibition. While long-term viability appears to be reduced by the over-expression of IRF4 itself, there is a significant abrogation of growth inhibition and a reduced induction of apoptosis over background in the presence of ectopic IRF4 after a 6-day incubation with CBP/EP300 inhibitor (*Figure 6A* (right) and *Figure 6—figure supplement 1A*).

If the IRF4-mediated suppression of MYC is required for the phenotypic effects of CBP/EP300 bromodomain inhibition, one would expect that ectopic expression of IRF4 should block MYC

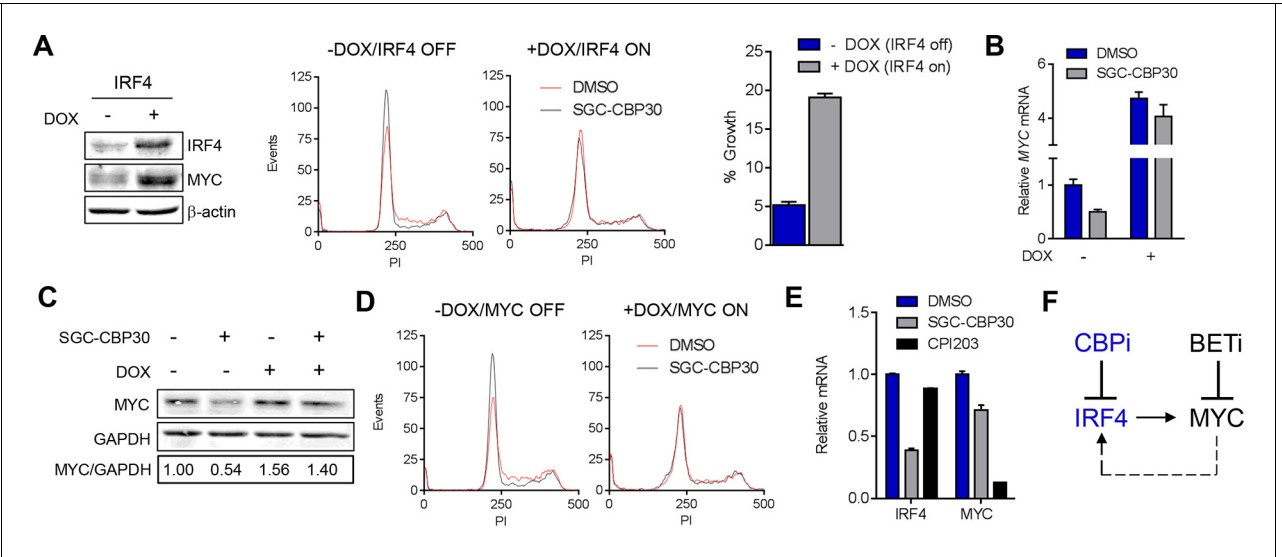

**Figure 6.** CBP/EP300 bromodomain inhibition suppresses the IRF4/MYC axis to cause viability defects. (**A**) IRF4 expression was induced in the LP1/IRF4 cell line by the addition of doxycycline. Left, lysates were prepared after 3 days and used for Western analysis with the indicated antibodies. Middle, cells were incubated for an additional 24 hr with DMSO or SGC-CBP30 (2.5 μM) and fixed for cell cycle analysis by flow cytometry. Representative histograms of two biological replicate experiments are shown. Right, Cells were incubated for 6 days in the presence of SGC-CBP30 (2.5 μM). Viable cells were counted by flow cytometry and percent growth was calculated relative to the DMSO-treated condition for induced or uninduced cells. Values represent the mean of n = 3 technical replicates, ± SEM (**B**) Cells were induced as in (**A**) and were treated with DMSO or SGC-CBP30 (2.5 μM) for 6 hr. Expression of *MYC* was measured by qRT-PCR, normalized to *GAPDH*, and expressed relative to uninduced cells treated with DMSO. Values represent the mean of n = 3 technical replicates, ± SEM. (**C**) As in (**B**) except cells were treated for 24 hr and lysed for Western analysis with the indicated antibodies. Values represent the ratio of GAPDH-normalized MYC expression relative to uninduced DMSO-treated cells. (**D**) MYC expression was induced in the LP1/MYC cell line by the addition of doxycycline. Cells were incubated for an additional 24 hr with DMSO or SGC-CBP30 (2.5 μM) and fixed for cell cycle analysis by flow cytometry. Representative histograms of two independent experiments are shown. (**E**) RNA sequencing data from *Figure 3A* is expressed as the mean of the two biological replicates ( ± SEM) normalized to DMSO-treated cells. (**F**) Model for the suppression of the IRF4/MYC axis by CBP/EP300 and BET bromodomain inhibitors.

The following figure supplement is available for figure 6:

**Figure supplement 1.** Additional data pertaining to IRF4 and MYC reconstitution experiments in *Figure 6.*

suppression. Indeed, we found that the induction of IRF4 in the LP-1/IRF4 cell line both increased *MYC* expression (most prominently at the mRNA level) and prevented the suppression of *MYC* by SGC-CBP30 and I-CBP112 (*Figure 6A* far left, 6B and 6C and *Figure 6—figure supplement 1E*). Consistent with MYC suppression being a critical downstream effect of IRF4 suppression, ectopic expression of MYC in the LP-1/MYC cell line phenocopied ectopic expression of IRF4, rescuing cell cycle arrest and abrogating MYC suppression following CBP/EP300 bromodomain inhibition (*Figure 6D* and *Figure 6—figure supplement 1B,C,D*, and 1F).

While BET proteins are known to similarly target MYC in multiple myeloma, a comparison of CBP/EP300 and BET bromodomain inhibition demonstrated that these modalities target the IRF4-MYC network at different nodes, with BET inhibition having no impact on IRF4 at the doses and timepoints examined (*Figure 6E*, *Figure 3D*, and *Figure 4—figure supplement 1B*). Our data suggest that CBP/EP300 bromodomain inhibition exerts its anti-myeloma effects in a mechanism distinct from BET inhibition via the direct transcriptional inhibition of *IRF4* and the downstream suppression of IRF4 target genes such as *MYC*.

## Discussion

In the current study, we demonstrate that CBP/EP300 bromodomain inhibition results in cell cycle arrest and apoptosis in multiple myeloma cell lines. Viability effects are dependent on the silencing of the transcription factor IRF4, which results in the downstream suppression of c-MYC. CBP/EP300

bromodomain inhibition thus targets the IRF4/MYC network, which is critical for multiple myeloma cells independent of the upstream oncogenic signal. A recent publication describes the use of the CBP/EP300 bromodomain inhibitor I-CBP112 to inhibit the growth of leukemic cells (*Picaud et al., 2015*). Our data pointing to the preferential activity of both SGC-CBP30 and I-CBP112 in multiple myeloma cell lines as compared to leukemic cell lines is consistent with this published work. Similar to our findings, Picaud et al. observed minor cytostatic and limited cytotoxic effects in all leukemic cell lines screened with the exception of Kasumi-1. Only upon examining the effects of I-CBP112 on clonogenic growth did the authors observe more broad phenotypic effects. Thus, while CBP/EP300 bromodomain inhibition may have robust cytotoxic effects in multiple myeloma, our results do not exclude the possibility that this modality would have additional therapeutic utility in leukemia by targeting leukemic self-renewal.

Pharmacological inhibition of CBP/EP300 bromodomains represents a viable strategy for targeting these transcriptional co-activators. Evidence from genetic studies in mice has shown that ablation of any two of the four alleles of CBP and EP300 results in embryonic lethality, and mouse embryonic fibroblasts lacking expression of CBP and EP300 cannot proliferate (*Kung et al., 2000*; *Jin et al., 2011*; *Yao et al., 1998*). The selective viability effects and limited transcriptional footprint observed with CBP/EP300 bromodomain inhibitors suggests that this modality is milder than genetic ablation, perhaps affording an acceptable therapeutic index once drug-like molecules are optimized. Our results are more broadly consistent with recent studies using SGC-CBP30 and I-CBP112 that demonstrated selective phenotypic and transcriptional effects in distinct biological contexts (*Hammitzsch et al., 2015*; *Picaud et al., 2015*).

Mice with heterozygous loss of *Cbp* are prone to the development of hematologic malignancies, and human patients with germline mutations in *CREBBP* develop the Rubinstein-Taybi cancer predisposition syndrome (*Kung et al., 2000*; *Iyer et al., 2004*). Further, recent surveys of the mutational landscape in a variety of tumors have demonstrated frequent loss of function mutations in *CREBBP* and *EP300* (*Pasqualucci et al., 2011*; *Peifer et al., 2012*; *Kishimoto et al., 2005*; *Mullighan et al., 2011*; *Zhang et al., 2012*). While this evidence implicates CBP/EP300 as tumor suppressors, evidence also supports their oncogenic activity. Rare human leukemias have been found with oncogenic fusion proteins containing either CBP or EP300, and these oncogenic fusion proteins require the activity of CBP or EP300 (*Murati et al., 2007*; *Ohnishi et al., 2008*; *Yung et al., 2011*; *Wang et al., 2011*). Genetic ablation and pharmacological inhibition of CBP/EP300 in leukemic cell lines and primary patient samples also support the oncogenic role of CBP and EP300 (*Picaud et al., 2015*; *Giotopoulos et al., 2015*). Our data in multiple myeloma are consistent with an activity supporting oncogenic signaling, as either pharmacological inhibition or knockdown resulted in loss of viability. It is unclear whether CBP/EP300 bromodomain inhibition would have tumor promoting activity in normal tissues. However, concerns about inhibiting potential tumor suppressor activity of CBP/EP300 in normal tissues may be alleviated by a dosing regimen that prevents continuous target coverage in normal tissues.

BET bromodomain inhibitors are highly active in hematologic malignancies, including multiple myeloma (*Delmore et al., 2011*; *Mertz et al., 2011*). The activity of CBP/EP300 bromodomain inhibitors in multiple myeloma potentially suggests that this modality may modify similar genes regulated by BET bromodomain inhibitors, but transcriptional profiling does not support this notion. At the doses of SGC-CBP30 utilized, CBP/EP300 bromodomain inhibition appears to have a more circumscribed transcriptional footprint than BET bromodomain inhibition. Phenotypic effects of BET bromodomain inhibition in multiple myeloma are likely due to direct suppression of *MYC* and *BCL2*, while the effects of CBP/EP300 bromodomain inhibition appear to be via suppression of *IRF4*. The distinct transcriptional effects of the two modalities suggests that combinations may be efficacious. It has in fact been shown that targeting the IRF4 network with lenalidomide and the MYC network with BET bromodomain inhibitors has synergistic effects in mantle cell lymphoma and primary effusion lymphoma (*Moros et al., 2014*; *Gopalakrishnan et al., 2015*). CBP/EP300 bromodomain inhibition may thus represent an alternative strategy for targeting the IRF4 transcriptional axis in these contexts.

The discovery of BET bromodomain inhibitors represented a breakthrough in the ability to target what were thought to be intractable oncogenic factors. Here we have shown that CBP/EP300 bromodomain inhibitors may similarly be used to target the expression of critical oncogenic transcription factors. As dysregulated transcriptional control is central to the pathology of cancer, the ability

to target oncogenic transcription networks with small molecule bromodomain inhibitors represents a promising direction for future therapeutics.

## Materials and methods

### Cell lines

Sources of cell lines and results of mycoplasma testing are provided as *Supplementary file 2*. All cell lines were used within 1–2 months of thawing from original stock vials received from supplier and were not further authenticated. LP-1 cells containing doxycycline-inducible IRF4 were generated as described for the inducible LP-1/MYC cell line (*Mertz et al., 2011*) using the IRF4 coding sequence (RefSeq BC015752.1) obtained from Origene Technologies, Inc. (Rockville, MD) as a template. Inducible cell lines were incubated with 1 µg/ml doxycycline (Sigma-Aldrich, Inc; St. Louis, MO) for 3 days. SGC-CBP30 (2.5 µM) or I-CBP112 (5 µM) was added for 6 hr for RNA analysis or for 24 hr, and cells were fixed for cell cycle analysis or pelleted for Western analysis, or were seeded in a 96 well plates for long -term viability testing.

### NanoBRET cellular assays

NanoBRET was carried out using the NanoBRET Protein:Protein Interaction System (Promega, Inc.; Madison, WI) according to the manufacturer's instructions. Briefly, HEK293 cells were transiently co-transfected with a vector for histone H3.3-HaloTag and a NanoLuc tagged CBP bromodomain expression construct. Transfected cells were plated in 96 -well plates in the presence or absence of ligand, then treated with dose titrations of indicated compounds. Readings were performed on an Envision Plate Reader (Perkin Elmer, Inc.; Waltham, MA) and BRET readings were calculated by dividing the acceptor emission value (600 nm) by the donor emission value (460/50 nm).

### Bromodomain chromatin release assay

As described previously (*Huang et al., 2014*), this assay monitors the compound-dependent release and aggregation of a fusion protein consisting of a bromodomain and the fluorescent protein ZsGreen. For the BRD4 chromatin release assay, U2OS cells capable of inducibly expressing the full-length BRD4 protein in fusion with ZsG were generated using the pLVT3G/ZsGreen-BRD4/TO3G vector and maintained in blasticidin at 15 µg/ml. Consistent with published data (*Dawson et al., 2011*), we did not observe global release of full length CBP fused to ZsGreen in response to compounds or bromodomain point mutations. Therefore, the bromodomain (BD) of CBP was cloned into full length BRD9 (replacing the BRD9 BD) in frame with a ZsGreen fluorescent tag (ZsG). U2OS cells capable of inducibly expressing the ZsGreen-CBPBD fusion protein were generated by lentiviral delivery of the pLVT3G/BRD9-ZsG-CBPBD/TO3G vector, which contains both the inducible fusion protein and the tet transactivator. Cells were selected and maintained in the presence of 15 µg/ml blasticidin. 5000 cells/well were seeded in 384-well imaging plates in the presence of 2 µg/ml doxycycline to induce the expression of ZsG-fusion proteins. After 16 hr of incubation with doxycycline, fresh media containing serial dilution of compounds were added to the cells for 2 hr at 37°C. Cells were fixed with 4% paraformaldehyde (PFA) dissolved in PBS for 15 min at room temperature. Images of cells were acquired using ImageXpress Micro (Molecular Devices, Inc.; Sunnyvale, CA) and processed with the Transfluor Module of MetaXpress software. Average pits per cell values were obtained from four adjacent images in each well with two technical replicates for each compound concentration. Dose-response curves were generated by plotting the average pits per cell values at each dose and $EC_{50}$ values were calculated by a four-parameter non-linear regression model in GraphPad Prism.

### Differential scanning fluorimetry

Differential scanning fluorimetry (DSF) was performed as described (*Niesen et al., 2007*) with the indicated bromodomains using the ViiA7 real time PCR system (Life Technologies, Inc.; Carlsbad, CA). Variable buffer compositions were used for the different bromodomain proteins with 12X SYPRO orange dye, 20 µM of the compounds or equivalent percentage of DMSO, and 4–8 µM of the indicated protein. A melting curve was established using a range of 25–95°C and a ramping rate of 3°C per minute. The melting temperature ($T_m$) for each sample was determined using the ViiA7

software (version 1.2.2) and the $\Delta T_m$ was calculated by subtracting the $T_m$ of the control from the $T_m$ of the compound treated sample.

## Cell cycle analysis and viability determination

Cells were plated at 5000–10000 cells per well of 96-well plates containing titrations of the compounds as indicated. After incubation, the cells were incubated with 500 µg/ml resazurin (Sigma) in PBS for 2–8 hr, and fluorescence was measured (Ex 530 nm, Em 590 nm). Cell cycle analysis was performed as described previously (*Mertz et al., 2011*). For visualization, DNA content histograms were generated with GraphPad Prism. Dose-response curves were generated by plotting the normalized percent growth, percent sub-G1 and percent increase in G0/G1 at each dose values. $GI_{50}$ values were determined as the concentration at which viability was 50% of the DMSO value and calculated by a four-parameter non-linear regression model in GraphPad Prism. Cell synchronization was performed as described (*Mertz et al., 2011*), and cells were released into media containing DMSO, 2.5 µM SGC-CBP30 or 5 µM I-CBP112.

## Lentiviral shRNA transduction

Lentiviral shRNA vector and packaging have been described previously (*Mertz et al., 2011*). Cells (2E6 cells/ml) were transduced with lentivirus at an MOI of 5–10 in 8 µg/ml sequebrene (Sigma) and centrifuged at 1000g for 2 hr. Cells were diluted to $1 \times 10^6$ cells/ml overnight. Infected cells were diluted to $2 \times 10^5$ cells/mL in 1 µg/ml puromycin and transferred to 96-well plates or TC flasks. After 3–4 days, cells in flasks were pelleted and used for qRT-PCR or Western analysis. Cells in 96-well plates were incubated for 9 days and fixed for cell cycle analysis, with passaging and fixing of aliquots as indicated. Target sequences for shRNAs were as follows: shEP300 1: 5'CGGAAACAG TGGCACGAAGAT3'; shEP300 2: 5'CGGAGGATATTTCAGAGTCTA3'; shEP300 3: 5'GCGGAATAC TACCACCTTCTA3'; shCBP 1: 5'CCTCTTTGGAGTCTGCATCCT3'; shCBP 2: 5'GAGCTTCCCAAG TTAAAGAAG3'; shCBP 3: 5'GCCCATTGTGCATCTTCACGA3'. For IRF4 knockdown, validated shRNA constructs were obtained from Sigma. Constructs shIRF4-1, shIRF4-2, shIRF4-3, shIRF4-4, and shIRF4-5 correspond to TRCN0000429523, TRCN0000014764, TRCN0000014765, TRCN0000014767, and TRCN000433892, respectively.

## mRNA sequencing, gene expression microarrays, data analysis, and quantitative RTPCR

Total RNA was prepared with an RNeasy Mini Kit (Qiagen, Inc.; Hilden, Germany) with on column DNAse digestion and submitted to Ocean Ridge Biosciences (Palm Beach Gardens, FL) for sequencing and mapping. The data in RPKM for each gene with compound treatment was compared to DMSO treatment, and $\log_2$ fold changes were used for further analysis. Rank ordered gene lists were used for Gene Set Enrichment Analysis (*Subramanian et al., 2005*). RNA preparation, cDNA synthesis, and qRT-PCR were performed as described (*Mertz et al., 2011*). For dose titration experiments, cells in 96 -well plates were lysed in lysis buffer (1% Triton X-100, 0.01 µM glycine pH 2.5) and used directly for cDNA synthesis and qRT-PCR. Primer sequences can be found in *Supplementary file 1*.

## Western analysis

Whole cell extracts were prepared by lysis in RIPA buffer + EDTA (Boston Bioproducts, Inc.; Ashland, MA) with protease inhibitor cocktail (Roche Life Sciences; Indianapolis, IN). Extracts were subjected to SDS-PAGE and Western analysis with MYC (Cell Signaling (Danvers, MA) #5605), IRF4 (Cell Signaling #4964 or #4948), GAPDH (Life Technologies AM4300), CBP (Santa Cruz (Dallas, TX) sc-369), EP300 (Santa Cruz sc-584), or β-actin (Life Technologies AM4302) primary antibodies. For histone analysis, extracts were prepared by sulfuric acid extraction of permeablized nuclei, and extracted histones were subjected to SDS-PAGE and Western analysis with H3K18ac (Cell Signaling #9675), H3K27ac (EMD Millipore (Billerica, MA) 07–360), or H4 (Abcam (Cambridge, MA) 31830). Blots were incubated with DyLight conjugated secondary antibodies, imaged and quantified with a Licor fluorescence imager (Licor, Inc.; Lincoln, NE), or with HRP- conjugated secondary antibodies for ECL visualization.

## ChIP-seq

$5 \times 10^7$ LP-1 cells were treated for 6 hr with DMSO or 2.5 µM SGC-CBP30 at a density of $5 \times 10^5$/ml. Cells were fixed with a final concentration of 1% formaldehyde for 10 min at room temperature. Glycine was added to a final concentration of 0.125 M to stop crosslinking. The cells were washed twice in cold PBS followed by lysis at 4°C for 1 hr in buffer containing 10 mM Tris-HCl pH 7.5, 10 mM NaCl, 5 mM MgCl$_2$, 0.2% NP-40, + protease inhibitor cocktail (Sigma). Following lysis, nuclei were recovered by centrifugation, and resuspended in buffer containing 10 mM Tris-HCl, 0.1 mM EDTA, 5 mM MgAc$_2$, 25% glycerol. An equal part 2X MNase buffer was added, containing 50 mM KCl, 8 mM MgCl$_2$, 2 mM CaCl$_2$, 100 mM Tris-HCl. Micrococcal nuclease (Roche) was added to 300 U/ml and chromatin was digested at room temperature for 20 min. Dilution buffer (0.1% SDS, 1.1% Triton-X 100, 2 mM EDTA, 20 mM Tris-HCl pH 8.0, 150 mM NaCl) was added and nuclei were broken down by sonication. Chromatin was cleared by centrifugation and pre-cleared with protein A-conjugated magnetic beads (Life Technologies). 10–20 µg pre-cleared chromatin was combined with 10 µg anti-EP300 antibody (Santa Cruz sc-585X) or 2.5 ug of anti-H3K18ac (Cell Signaling, 9675) or anti-H3K27ac (Abcam, ab4729) conjugated to protein A magnetic beads. IPs were performed overnight at 4°C. Immune complexes were washed twice in buffer containing 140 mM NaCl, once in buffer containing 360 mM NaCl, once in 250 mM LiCl wash buffer, and twice in TE. Samples were eluted and treated with 20 µg proteinase K (Roche) for 1 hr at 55°C, crosslinks were reversed for 4 hr at 65°C, and 20 µg RNase (Sigma) was added for 1 hr at 37°C. DNA was purified with the MinElute kit (Qiagen), and libraries were prepared using the Ovation Ultralow DR Multiplex System (NuGEN, Inc.; San Carlos, CA) according to the manufacturer's recommendations. Amplified libraries were size selected and gel-purified prior to Illumina massively parallel sequencing on a HiSeq 2000 system at the MIT Biomicro Center. Biological replicates were performed for each sample, and representative images are depicted.

## AlphaLISA

Inhibitory activity of compounds was determined by following the inhibition of the binding of purified His-Flag-tagged bromodomains to H4-TetraAc-biotin peptide (New England Peptide, Inc.; Gardner, MA) using AlphaLISA technology (Perkin Elmer). Compound at varying concentrations were dispensed into 384 well Proxiplates (Perkin Elmer) using Echo technology (Labcyte, Inc.; Sunnyvale, CA). For CBP assays, 0.5 µM His-Flag-tagged CBP bromodomain (amino acids 1082–1197) was incubated with 0.003 µM H4-TetraAc-biotin for 20 min at room temperature in 1x reaction buffer (50 mM HEPES pH 7.5, 1 mM TCEP, 0.069 mM Brij-35, 150 mM NaCl, and 0.1 mg/ml BSA). Streptavidin acceptor beads and nickel donor beads (Perkin Elmer) were added to 15 µg/ml with a Combi Multidrop dispenser. Plates were sealed and incubated at 90 min in the dark at room temperature, and plates were read on an Envision plate reader (Perkin Elmer) according to manufacturer's instructions. For the BET assays, the protocol was similar except that BET family bromodomains were used at 0.03 µM (BRD4-BD1) and incubated with 0.2 µM H4-TetraAc-biotin for 20 min in reaction buffer (40 mM HEPES pH7.0, 1 mM DTT, 0.069 mM Brij-35, 40 mM NaCl, and 0.1 mg/ml BSA). Streptavidin donor beads and Anti-Flag Acceptor beads (Perkin Elmer) were added to 10 µg/ml, and then plates were sealed and incubated in the dark for 60 min prior to reading on the Envision.

## Chemical compounds

The synthesis and characterization of CPI203 have been published previously (*Devaiah et al., 2012*). SGC-CBP30 and I-CBP112 are commercially available (Sigma).

## Acknowledgements

We thank the many Constellation and Genentech employees in their support of these studies. Special thanks to Prerna Kotak, Ted Peters, and Gina Prophete for technical support and Jim Audia, Patrick Trojer, Keith Dionne, Jeff Settleman, Nicole Follmer, Jose Lora, Richard Cummings, Michael Cooper, JC Harmange, Brian Albrecht, and David Stokoe for helpful discussions and comments on the manuscript.

## Additional information

### Competing interests

ARC, RCC, AN, PJK, SJ, KLS, PS, CH, EP, LZ, AB-R, BMB, JAM, RJS: Employee of Constellation Pharmaceuticals. KEG: Employee of Genentech.

### Funding

No external funding was received for this work.

### Author contributions

ARC, RCC, Conception and design, Acquisition of data, Analysis and interpretation of data, Drafting or revising the article; AN, SJ, KLS, EP, AB-R, Acquisition of data, Drafting or revising the article; PJK, PS, Acquisition of data, Analysis and interpretation of data, Drafting or revising the article; CH, LZ, BMB, Analysis and interpretation of data, Drafting or revising the article; KEG, Conception and design, Drafting or revising the article; JAM, RJS, Conception and design, Analysis and interpretation of data, Drafting or revising the article

### Author ORCIDs

Adrianne Neiss, http://orcid.org/0000-0001-5329-782X

## Additional files

### Supplementary files

• Supplementary file 1. Includes qPCR primer sequences and UPL probe numbers for RT-qPCR experiments described in the manuscript.

• Supplementary file 2. Indicates the source of all cell lines used, as well as results of mycoplasma testing throughout the course of these studies.

### Major datasets

The following datasets were generated:

| Author(s) | Year | Dataset title | Dataset URL | Database, license, and accessibility information |
|---|---|---|---|---|
| Conery AR, Centore RC, Hatton C, Bryant B, Sims III RJ | 2015 | Bromodomain inhibition of the transcriptional coactivators CBP/EP300 as a therapeutic strategy to target the IRF4 network in multiple myeloma | https://www.ncbi.nlm.nih.gov/geo/query/acc.cgi?acc=GSE71911 | Publicly available at NCBI Gene Expression Omnibus (Accession no: GSE71911). |

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
