## [Decision Letter]

Thank you for submitting your work entitled "CBP/EP300 bromodomain inhibition as a therapeutic strategy to target the IRF4 network in multiple myeloma" for peer review at *eLife*. Your submission has been favorably evaluated by Charles Sawyers (Senior editor) and three reviewers, one of whom is a member of our Board of Reviewing Editors.

The reviewers have discussed the reviews with one another and the Reviewing editor has drafted this decision to help you prepare a revised submission.

Summary:

Chemical inhibitors of CBP/P300 bromodomain are evaluated in hematopoietic cancer cell lines for effects on proliferation and gene expression. After validating the selectivity of these compounds in biochemical and in-cell assays for P300/CBP versus BRD4, the sensitivity of a large panel human myeloma and leukemia cell lines to these compounds was evaluated. This revealed sensitivity within a variety of myeloma cell lines in the low micromolar range of GI50s. Comparative transcriptional profiling revealed a unique IRF4 gene signature and ensuing genetic experiments provided evidence supporting IRF4 and MYC suppression as an important effect of these compounds.

Essential revisions:

1) The manuscript is of interest, but given that these are previously published compounds and the mechanistic studies are not particularly detailed, there should be more support for the proposed pathway IRF4/MYC as the critical pathway downstream of P300/CBP Bromodomain inhibition and the specificity of the compounds for P300/CBP.

2) Figure 4 and Figure 5 show data using CBP30 to inhibit CBP bromodomains and it is explained that this molecule was chosen since it is most potent. It seems reasonable to characterize this compound in greatest detail. However it would be helpful if they could assess the other less potent compound in some of the assays performed in Figure 3 to show similar effects.

3) In Figure 2 it appears that the compound induces significant apoptosis (sub-G1 cells). Is this rescued but IRF4 expression in the experiments described in Figure 5? The experiments in this figure are important but complicated by the toxicity of IRF4 itself. Along these lines, the authors argue that MYC maybe downstream of IRF4. Does Myc expression rescue some of the phenotypes induced by the inhibitor? In general it would be helpful if there were some more evidence that IRF4 and MYC are critical to the phenotypes. It appears they might be but this part of the manuscript is under developed.

4) Can the authors use ChIP-qPCR at the IRF4 super-enhancer and a few other regions to compare occupancy of BRD4, p300, and CBP with a titration of BET inhibitor and CBP/p300 bromodomain inhibitors. The goal would be to establish clarity whether the CBP/p300 compounds are in fact not influencing BRD4 at the concentrations of compound used here.

5) The authors have not clearly delineated the specificity of these agents. If activity data across the entire family of bromodomains is available this should be provided. If not, the compounds should be profiled in something like the commercially available Bromoscan. In the absence of such data, it is not clear whether effects on other bromodomains may contribute to the phenotypes observed.

6) The rationale for using a BRD9 – CBP bromodomain fusion protein for the chromatin release assay is entirely unclear and may substantially alter the effects of the compound on CBP occupancy which may or may not have other chromatin tethering capabilities. This experiment should be done with CBP and/or P300.

---

## [Author Response]

*1) The manuscript is of interest, but given that these are previously published compounds and the mechanistic studies are not particularly detailed, there should be more support for the proposed pathway IRF4/MYC as the critical pathway downstream of P300/CBP Bromodomain inhibition and the specificity of the compounds for P300/CBP.*

We thank the reviewers for these suggestions, as we feel that the additional data in the revised version of the manuscript substantially strengthens the connection between CBP/EP300 bromodomain inhibition and the IRF4/MYC axis. Regarding the selectivity of the compounds, we have added data in [Supplementary-material SD1-data] showing Differential Scanning Fluorimetry (DSF) profiling of the compounds across 19 members of the bromodomain family, supporting the specificity towards CBP. These data are consistent with those provided in two manuscripts that were published during the revision period (Hammitzsch et al., Proc. Natl. Acad Sci. USA 2015, Picaud et al. Cancer Res. 2015), and together with our additional selectivity data argue that the effects we have observed are due to on-target engagement of CBP/EP300 bromodomains. We have also generated additional supporting data with I-CBP112 to demonstrate that the effects we have observed are not specific to any given chemotype.

Regarding the connection to the IRF4/MYC axis, we have generated several new pieces of data to support our hypotheses. First, we show in Figure 5 that the pharmacodynamic response of IRF4 suppression is correlated with in vitro efficacy of SGC-CBP30. Second, we used IRF4 knockdown experiments to demonstrate that cell lines that are phenotypically sensitive to CBP/EP300 bromodomain inhibition require IRF4 for viability (Figure 5 and Figure 5—figure supplement 1). The degree of MYC suppression resulting from IRF4 knockdown or CBP/EP300 bromodomain inhibition is correlated with the phenotypic response to either stimulus (Figure 5 and Figure 5—figure supplement 1), which lends further support to MYC being a critical downstream factor of IRF4 in multiple myeloma. Third, we have shown in an additional cell line, OPM2, that CBP/EP300 knockdown leads to IRF4 and MYC suppression (Figure 5—figure supplement 1). Finally, we have used reconstitution with ectopic MYC to further support the hypothesis that MYC suppression downstream of IRF4 suppression is a critical factor in the phenotypic response to CBP/EP300 bromodomain inhibition (Figure 6 and Figure 6—figure supplement 1).

*2) Figure 4 and Figure 5 show data using CBP30 to inhibit CBP bromodomains and it is explained that this molecule was chosen since it is most potent. It seems reasonable to characterize this compound in greatest detail. However it would be helpful if they could assess the other less potent compound in some of the assays performed in Figure 3to show similar effects.*

Where possible, we have replicated experiments with I-CBP112 in the revised manuscript. Figure 2—figure supplement 1 demonstrates a consistent phenotypic response with the two compounds. Figure 3—figure supplement 1 shows a similar transcriptional response for the two compounds, as well as a similar distinction from the effects of CPI203. Figure 4—figure supplement 1 and 1D show suppression of *IRF4* mRNA and protein with I-CBP112, while Figure 4—figure supplement 1 show similar kinetics of *IRF4* suppression with the two compounds. Figure 5—figure supplement 1 shows dose dependent reduction in *MYC* mRNA with I-CBP112 in two cell lines. Finally, Figure 6—figure supplement 1 and 1F demonstrate IRF4 and MYC reconstitution with I-CBP112 treatment. These data support our conclusions with SGC-CBP30 and argue that the effects we have observed are not scaffold-specific.

*3) In Figure 2 it appears that the compound induces significant apoptosis (sub-G1 cells). Is this rescued but IRF4 expression in the experiments described in Figure 5? The experiments in this figure are important but complicated by the toxicity of IRF4 itself. Along these lines, the authors argue that MYC maybe downstream of IRF4. Does Myc expression rescue some of the phenotypes induced by the inhibitor? In general it would be helpful if there were some more evidence that IRF4 and MYC are critical to the phenotypes. It appears they might be but this part of the manuscript is under developed.*

Regarding the first point, the reviewers are correct in pointing out that ectopic expression of IRF4 leads to increased basal apoptosis (we believe this is due to increased MYC expression, which is known to promote apoptosis). However, we have shown in Figure 6—figure supplement 1 that the increase in apoptosis (measured by% sub-G1 cells, in response to either SGC-CBP30 or I-CBP112) above background is reduced in the presence of ectopic IRF4. As mentioned above, we have added experiments with MYC reconstitution to the revised manuscript, and demonstrate that ectopic expression of MYC phenocopies ectopic expression of IRF4 in this context. Additional evidence connecting suppression of IRF4/MYC with the phenotypic response to CBP/EP300 bromodomain inhibitors is discussed in point 1 above.

*4) Can the authors use ChIP-qPCR at the IRF4 super-enhancer and a few other regions to compare occupancy of BRD4, p300, and CBP with a titration of BET inhibitor and CBP/p300 bromodomain inhibitors. The goal would be to establish clarity whether the CBP/p300 compounds are in fact not influencing BRD4 at the concentrations of compound used here.*

We thank the reviewers for this suggestion, as it further addresses the specificity of SGC-CBP30 towards CBP/EP300 bromodomains. In our experience, manual ChIP-qPCR has been difficult to interpret when using small changes in inhibitor concentrations (i.e. dose-responses), perhaps reflecting inherent assay noise. To address this question, we performed genome-wide ChIP-seq with BRD4 antibodies to address the global changes in BRD4 occupancy following CBP/EP300 bromodomain inhibition. We utilized the same concentrations of inhibitors used in gene expression and functional experiments in the manuscript for consistency (6 hour treatment with 2.5 μM SGC-CBP30, 0.25 μM CPI203). We show in Figure 4—figure supplement 2 that while BRD4 is evicted from some loci upon treatment with SGC-CBP30, there is little correlation with the effects of the pure BET inhibitor CPI203. In contrast, in Figure 4—figure supplement 2 we show that SGC-CBP30 leads to eviction of EP300 at numerous loci, an effect that is not observed with CPI203 (which in fact increases EP300 enrichment at multiple loci). We finally show some example loci with BRD4 and EP300 ChIP-seq tracks following SGC-CBP30 and CPI203 treatment (Figure 4—figure supplement 2). Note the eviction of EP300 at the IRF4 super-enhancer and at the ARHGEF12 locus (which is accompanied by minimal eviction of BRD4) in response to SGC-CBP30. In contrast note the robust eviction of only BRD4 at the IgH enhancer (which drives *MYC* in this cell line) and loci such as CLECL1 in response to CPI203. These data argue that the effects of SGC-CBP30 are not due to global eviction of BRD4, which is observed only with the pure BET inhibitor CPI203.

*5) The authors have not clearly delineated the specificity of these agents. If activity data across the entire family of bromodomains is available this should be provided. If not, the compounds should be profiled in something like the commercially available Bromoscan. In the absence of such data, it is not clear whether effects on other bromodomains may contribute to the phenotypes observed.*

As discussed in point 1 above, we have added DSF data across a panel of bromodomains to the revised manuscript ([Supplementary-material SD1-data]). We further reference the recently published manuscripts from the SGC to support our data.

*6) The rationale for using a BRD9* – *CBP bromodomain fusion protein for the chromatin release assay is entirely unclear and may substantially alter the effects of the compound on CBP occupancy which may or may not have other chromatin tethering capabilities. This experiment should be done with CBP and/or P300.*

We thank the reviewers for pointing out the ambiguity in our explanation for making use of the BRD9 scaffold to test the CBP bromodomain. We have added additional text addressing the assay to the Methods section so as not to distract the reader in the Results section. Consistent with data published in Picaud et al. (Cancer Res. 2015) we did not observe consistent release of chromatin-bound full-length CBP-ZsGreen fusion proteins in response to pharmacologic inhibition or inactivating mutation of the bromodomain. The group at the SGC rectified this problem by fusing three copies of the CBP bromodomain to GFP for FRAP experiments. Our solution was to swap the bromodomain of CBP into the BRD9 scaffold. We found that this domain-swapped fusion protein was released from chromatin only by compounds that showed biochemical engagement of CBP, and that swapping distinct bromodomains into the BRD9 scaffold allowed for chromatin release only with compounds that targeted that specific bromodomain (see our published patent application WO/2014/144303 for reference). While we cannot completely rule out confounding effects of using the chimeric CBP bromodomain in the BRD9 scaffold, our data with the chromatin release assay is complimentary to the BRET assay in Figure 1, as well as to the published cellular target engagement assays for SGC-CBP30 and I-CBP112 from the SGC.